# Effects of Scallop Mantle Toxin on Intestinal Microflora and Intestinal Barrier Function in Mice

**DOI:** 10.3390/toxins16060247

**Published:** 2024-05-27

**Authors:** Xiong Geng, Ran Lin, Yasushi Hasegawa, Luomeng Chao, Huayan Shang, Jingjing Yang, Weina Tian, Wenting Ma, Miaomiao Zhuang, Jianrong Li

**Affiliations:** 1College of Food Science and Engineering, Bohai University, Jinzhou 121013, China; gengxiong@qymail.bhu.edu.cn (X.G.); yingying2011510@163.com (R.L.); shy2023015150@bhu.edu.cn (H.S.); kjcy10@163.com (W.T.); mwt2024156027@126.com (W.M.); zmm13865981106@163.com (M.Z.); 2College of Environmental Technology, Muroran Institute of Technology, Muroran 050-8585, Japan; hasegawa@muroran-it.ac.jp; 3College of Animal Science and Technology, Inner Mongolia Minzu University, Tongliao 028000, China; chaoluomeng@imun.edu.cn; 4Kerqin District Testing Institute for Food and Drug Control, Tongliao 028000, China; xiaojingyang.hi@163.com

**Keywords:** scallop mantle, proteotoxin, gut microbiota, intestinal barrier

## Abstract

Previous studies have shown that feeding mice with food containing mantle tissue from Japanese scallops results in aggravated liver and kidney damage, ultimately resulting in mortality within weeks. The aim of this study is to evaluate the toxicity of scallop mantle in China’s coastal areas and explore the impact of scallop mantle toxins (SMT) on intestinal barrier integrity and gut microbiota in mice. The Illumina MiSeq sequencing of V3-V4 hypervariable regions of 16S ribosomal RNA was employed to study the alterations in gut microbiota in the feces of SMT mice. The results showed that intestinal flora abundance and diversity in the SMT group were decreased. Compared with the control group, significant increases were observed in serum indexes related to liver, intestine, inflammation, and kidney functions among SMT-exposed mice. Accompanied by varying degrees of tissue damage observed within these organs, the beneficial bacteria of *Muribaculaceae* and *Marinifilaceae* significantly reduced, while the harmful bacteria of *Enterobacteriaceae* and *Helicobacter* were significantly increased. Taken together, this article elucidates the inflammation and glucose metabolism disorder caused by scallop mantle toxin in mice from the angle of gut microbiota and metabolism. SMT can destroy the equilibrium of intestinal flora and damage the intestinal mucosal barrier, which leads to glucose metabolism disorder and intestinal dysfunction and may ultimately bring about systemic toxicity.

## 1. Introduction

By 2022, China will produce more than 20 million tons of mariculture aquatic products, including nearly 16 million tons of shellfish and nearly 1.8 million tons of scallops. There is a wide variety of scallops, with about 45 species distributed along China’s coasts, among which the most common and important economic scallops are *Chlamys farreri*, *Bay scallops*, and *Patinopecten yessoensis*. Scallops have long been one of the important species of mariculture in China. The farmed output of marine shellfish accounts for 90.79% of the world’s marine shellfish output. The top 10 producers are China, the United States, Japan, South Korea, Chile, Spain, Thailand, France, Canada, and Italy, accounting for 94.17% of the world’s marine shellfish output [1]. China’s scallop output mainly comes from breeding, with a small proportion of fishing. In 2021, China imported USD 253 million scallops in 2021 and exported USD 305 million.

Shellfish processing can be divided into purification, pretreatment, deep processing, and waste treatment according to the technological process. The finished products mainly include fresh products, frozen products, dry products, cured and smoked products, tank products, additives, condiments, small packaging leisure products, and medical products [2]. Scallops are an important seafood in Hokkaido, Japan. The scallop mantle in Japan is commonly consumed in various forms, such as when raw, smoked, dried, barbecued, frozen, and boiled. Recent studies have revealed that the scallop mantle possesses a rich content of aminoglyaccharides, unsaturated fatty acids, active peptides, taurine, and other bioactive substances. These components exhibit diverse physiological functions, including anti-aging properties, anti-tumor effects, antioxidant activity, blood lipid reduction abilities, and immune regulatory capabilities [3]. In order to improve the added value of shellfish products, Qin Yu et al. [4] studied the process of preparing seafood soy sauce by fermentation in 2021, using scallop mantle as the main raw material. In 2013, Wanghui Qing et al. [5] developed scallop mantle sausage with a unique seafood flavor using scallop mantle and chicken as raw materials. Through investigation, it is found that there are many scallop mantle processed foods in the Chinese market, which are mainly sold in the form of dry products, smoked products, ready-to-eat snacks, canned products and hoisin sauce, etc. In certain coastal regions, the consumption of raw or cooked dishes prepared with scallop mantle as a primary or ancillary ingredient is prevalent among the local population [6].

In 2018, Yasushi Hasegawa [7] of Japan first identified a new kind of shellfish toxin, which is distinct from DSP and PSP toxin in *Patinopecten yessoensis* scallops after feeding scallop mantle to rats; the rats lost their appetite and died a few weeks later. So far, Hasegawa et al. have isolated and identified this new shellfish poison and characterized it as a proteotoxin capable of inducing liver and kidney damage in rats. It has been observed that this toxin exhibits remarkable stability within mantle tissues, particularly when exposed to acidic conditions and digestive enzymes, rendering it resistant to decomposition even at high temperatures. They conducted acute toxicity tests on the mantle tissue. Rats fed with 20% mantle tissue did not have significantly increased toxicity compared to rats fed with 1% mantle tissue, indicating that raising mantle tissue did not have acute toxicity. Studied the mantle tissue toxicity of small intestine wall tissue. Long-term feeding on mantle tissue can alter the color of the small intestine in rats. Real-time polymerase chain reaction analysis showed that the uptake of mantle tissue caused changes in the small intestine’s inflammation and endoplasmic reticulum stress markers. These outcomes indicate that scallop mantle feeding leads to toxicity after initial damage to the small intestinal tissue [8].

The gut microbiota is a complex microbial community that inhabits the digestive tract of animals and plays a significant role in host metabolism [9], nutrient absorption or production, and the immune system, making a significant contribution to the wellness of the parasite. The small intestine serves as the primary barrier to prevent pathogens and toxins from entering the human body [10], maintaining the steady state of intestinal flora and safeguarding organ integrity. The intestinal barrier integrity damage or defects may lead to bacteria imbalance and hazardous substances through the epithelial barrier, leading to intestinal inflammatory. The intestinal microbiota plays a significant role in mediating interactions between parasite metabolism and environmental materials.

After entering the human body, fungal toxins first interact with the gastrointestinal tract. The synergistic effect between the gut microbiota and the gastrointestinal tract can protect the parasite from fungal toxins. However, the specific changes of the individual intestinal microbiota disturbed by SMT are still unclear. *Patinopecten yessoensis* is one of the main aquatic products in the northeast region of China. Besides the adductor muscle, scallop mantle tissue is consumed in China. We must ensure the health and safety of human life and remove the new shellfish toxins in the scallop mantle so that humans can eat scallop products. This article aims to investigate the presence of a new type of shellfish toxin on scallop mantle in Dalian, Liaoning Province, China, and the effects of this toxin on the liver, kidney, and intestine of mice and to study the effects of a new type of shellfish toxin on intestinal flora and intestinal injury. This paper elucidates the toxic mechanism of scallop toxin from the perspective of intestinal flora, which provides a new idea for scientific, reasonable, and rapid detection technology and detoxification and also provides important theoretical and practical support for the safety of scallop mantle processing and consumption.

## 2. Results and Discussion

### 2.1. Changes in Body Weight and Food Intake

To examine the subacute oral toxicity of the mantle from China, SPF mice were fed a diet including 2% mantle. There were no discernible differences in overall appearance or behavior between the SMT and CON groups of mice, and no clinical symptoms were discovered within 2 weeks after commencing the mantle diet. Nevertheless, after 3 weeks, the food intake of the mice in the mantle diet group began to decrease significantly, and the body weight of the mice decreased accordingly. After a duration of 4 weeks, the mice on the mantle diet exhibited a reduction in food intake to approximately 40% of their initial consumption (Figure 1b). Furthermore, their body weight experienced a decline to around 60% of the average weight observed in the SMT group (Figure 1a). Prolonged consumption of the mantle diet for an additional period of 4–5 weeks resulted in symptoms such as dyspnea, abnormal movement patterns, a tendency toward lying behavior, tremors, decreased responsiveness, lethargy, weakness, and wasting. The mortality rate was notably high among mice in this stage experiment. The mice died at 5 weeks after administration, and the organs were measured as a percentage of body weight (Table 1). There was an important increase in kidney weight in the mantle diet group, while there was no important difference in liver, stomach, or intestinal weight between the control and mantle diet groups. This suggests that scallop coats in coastal regions of China also contain toxins. In order to further verify whether the toxins contained were the same as those in Japan, further liver and kidney toxicity tests were performed.

### 2.2. Effects of Scallop Mantle Toxin on Liver Damage

Liver serum markers and histopathology were analyzed to test for mantle epithelial layer toxicity. We measured liver serum markers in mice: AST, ALT, TG, TC, γ-GTP, and TBA. Mice fed the mice mantle had significantly higher vitality of ALT, AST, TC, γ-GTP, TG, and TBA (*p* > 0.05). The liver is the only organ in the human body without pain-sensing nerves [11], and it is easily destroyed by toxic chemicals, leading to metabolic dysfunction [12]. Under the stimulation of alcohol metabolism, oxidative stress occurs in the liver, causing damage to liver cells. ALT and AST (Figure 2c,d) are the most direct and sensitive indicators to reflect the damage to liver cells. Dong et al. [13] found that ALT and AST are high in healthy liver but low in blood. Once liver cells are damaged, a large amount of ALT and AST will be released into the serum, raising the serum level. ALT and AST levels reflect the damage degree of liver cells in a certain range [14]. Reddy et al. [15] point out that TG and TC levels (Figure 2e,f) in serum would increase when liver lipid metabolism was impaired. When hepatocellular disease or intra- and extra-hepatic obstruction occurs, the metabolism of bile acid is obstructed, and the serum total bile acid concentration is increased. Therefore, changes in TBA (Figure 2h) levels can sensitively reflect liver function. Elevated serum γ-GTP levels (Figure 2g) suggest liver injury. In the control group (Figure 2a), the structure of hepatic lobules and nuclei was clear, hepatic lobules were normal, there was no obvious congestion in hepatic sinuses, and no inflammatory cell infiltration was visible in liver cells. The liver tissue of the scallop mantle group (Figure 2b) mice showed obvious cell shrinkage and unclear boundary of hepatic lobules. The nuclear structure was seriously broken, and evident inflammatory cell infiltration was observed in the liver cells.

Therefore, we have observed that the consumption of scallop mantle can induce liver toxicity and damage in mice. The patterns of liver tissue injury align with those of Yasushi Hasegawa et al. [7].

### 2.3. Effects of Scallop Mantle Toxin on Nephridial Tissue Damage

In order to examine the toxicity of the mantle, we found the kidney serum markers and histopathology. CRE and BUN in serum are important indexes to evaluate kidney function; BUN is the main product of metabolism in the body. Its level will elevate when renal insufficiency occurs, and it can be used as a marker for early identification of kidney injury. CRE can accurately reflect the glomerular filtration function, and the level of CRE increases during renal injury. As can be seen from the picture (Figure 3c,d), serum CRE and BUN contents in the scallop mantle group were significantly increased compared with the CON group (*p* > 0.05). The results indicated that the mice of the scallop mantle group had kidney damage. In the control group (Figure 3a), the structure of glomeruli and renal tubules was clear and complete, and there was no abnormal morphological structure. Compared with the CON group, the kidney tissues of mice in the scallop mantle group (Figure 3b) exhibited obvious pathological changes, which were mainly manifested as glomerular contraction malformation, tubular dilatation and necrosis, severe expansion of proximal and distal curved tubules, and inflammatory cell infiltration in renal interstitium. Therefore, we discover that a feed diet containing the scallop mantle can produce toxic effects on the kidneys of mice, causing injury. The results of renal tissue injury were similar to those of Yasushi Hasegawa et al.

### 2.4. Effects of Scallop Mantle Toxin on Intestinal Tissue Damage

In the control group (Figure 4a), the jejunum mucosa was intact, the villi were arranged neatly, and the intestinal epithelial cells were of clear shape, densely arranged, and of uniform size. No obvious pathological changes were observed. In comparison with the CON group, damage to the intestinal barrier and structural integrity of the jejunum was obvious in the SMT group. There was significant inflammatory cytokine infiltration, and the villi height was also reduced (Figure 4c). As is shown in the figures (Figure 4d,f), serum concentrations of LPS, IL-6, and TNF-α of mice in the SMT group were significantly increased compared with those in the CON group, TNF-α, IL-6, and LPS are measures of pro-inflammatory cytokines, and enhanced serum concentrations indicate an inflammatory response in the body. An intact intestinal structure is essential for maintaining intestinal barrier function.

After eating the diet consisting of scallop mantle, the intestinal structure of mice experienced severe damage, which led to the breakdown of intestinal barrier function and systemic inflammation.

### 2.5. Effect of Scallop Mantle Toxin on the Alpha and Beta Diversity

By high-through sequencing analysis of 16S rDNA genes isolated from feces of bacteria, the effect of scallop mantle toxin on mice intestinal flora was studied [16]. Alpha diversity is utilized for assessing the microbial community diversity within a given sample. By calculating the values of each sample, the diversity index can provide insights into the richness, diversity, and uniformity of microbial communities in the sample.

As shown in Figure 5a,b, the SMT group had no significant impact on the Shanon and Chao1 indexes of fecal intestinal flora α-diversity compared with the control group (*p* > 0.05). Additionally, the CON and SMT groups had 1031 and 951 OTUs, respectively, with a total of 899 common operating taxonomic units (OTUs) between the two groups (Figure 5c). These results indicated that the SMT group was less than those of the CON group, suggesting that the bacterial community richness decreased due to SMT. In addition, in order to understand the effect of SMT on the intestinal microflora profile of mice, principal component analysis (PCA) of Bray–Curtis distance was performed based on OTU. The SMT group showed a significant deviation from the control group, revealing another mode (Figure 5d). The result further showed that the composition of intestinal flora in SMT intervention mice has undergone significant changes.

### 2.6. Effect of Scallop Mantle Toxin on Intestinal Microbial Composition

The dominant species influence the ecological and functional structure of the microbial community to a great extent. Comprehending the species composition of communities at various levels may effectively explain the formation, change, and ecological effect of community structure [17]. We selected the top 20 species based on their average abundance ranking from samples for demonstration.

At the phylum level (Figure 6a), there is the *Bacteroidota* (58.19%), *Firmicutes* (28.92%), and *Verrucomicrobiota* (7.77%) in the control group and the main *Bacteroidota* (45.34%) and *Firmicutes* (27.05%) and *Proteobacteria* (12.01%) in the scollop mantle group. The abundance of *Proteobacteria* in the SMT group increased compared with that of the CON group. The gut microbiome interacts with the host to maintain homeostasis, and imbalances can lead to disease. *Firmicutes* and *Bacteroidetes* (F/B) are the most diverse phyla that affect host physiology in humans and mice; the imbalanced F/B ratio is related to different disease processes [18]. Compared with the control group, the F/B of the scallop mantle group increased, indicating an imbalance in the proportion of intestinal flora in the SMT group. We speculated that the intake of scallop mantle promoted an increase in F/B, indicating that scallop mantle destroyed the structure of intestinal flora and exacerbated metabolic disorders. Proteobacteria are considered indicators of ecological imbalance and disease risk.

Following the consumption of scallop mantle, the increased abundance of *Proteobacteria* and faster growth of intestinal pathogens may be associated with the ingestion of bacteriocins produced by scallop mantle. At the family level (Figure 6b), compared with the control group, the abundance of the beneficial bacterium *Muribaculaceae* significantly decreased, while the abundance of harmful bacterium *Enterobacteriaceae* significantly increased.

The richness of *Marinifilaceae* decreased from 6.64% to 3.15%. *Marinifilaceae* is a family of beneficial bacteria associated with the treatment and improvement of lipid metabolism disorders [19]. *Enterobacteriaceae* is the Gram-negative family that can bring about various diseases in humans and animals, for example, osteomyelitis, urinary tract infections, and bacteremia.

At the genus level (Figure 6c), the main *Muribaculaceae* (23.02%), *Bacteroides* (11.80%), *Akkermansia* (7.77%), and *Odoribacter* (6.42%) are in the control group. In the SMT group are the main *Bacteroides* (14.86%), *Muribaculaceae* (11.14%), *Akkermansia* (10.70%), *Escherichia-Shigella* (10.30%). Compared with the control group, the abundance of beneficial bacteria Muribaculaceae decreased significantly. Liang H et al. [20] conducted an experiment in which metformin-regulated intestinal microbiota reduced liver damage associated with sepsis. After the application of metformin to rat liver injury, it was found that the proportion of *Muribaculaceae* increased, inferring that *Muribaculaceae* may be related to the treatment and improvement of steps-related liver injury. Changes in *Muribaculaceae* richness can result in a loss of intestinal barrier completeness; therefore, it increases the risk of cancer. The regulation of the abundance of *Muribaculaceae* could provide the foundation for ameliorating inflammation and metabolic diseases such as obesity. Therefore, we assumed that the decrease in the abundance of *Muribaculaceae* in scallop mantle led to the loss of intestinal barrier integrity, resulting in intestinal barrier damage.

In comparison with the CON group, the abundance of harmful bacteria *Escherichia-Shigella* increased significantly. Its increase causes the homeostasis of the intestinal flora to be disrupted, resulting in diarrhea and gastrointestinal diseases. The richness of *Blautia* decreased from 2.07% to 0.62% in the SMT group compared with the control group. *Blautia* is a genus of anaerobic bacteria with probiotic characteristics, widely present in the feces and intestines of mammals; it possesses the capacity to regulate host health, alleviate metabolic syndrome, and mitigate metabolic and inflammatory diseases. A reduction in its abundance may result in health complications such as inflammation and disrupted metabolism within the host.

Compared with the control group, the harmful *Helicobacter* abundance increased from 0.56% to 1.96% in the SMT group. *Helicobacter* causes varying degrees of acute pathology, ranging from gastroenteritis to chronic pathology, including inflammatory bowel disease and liver and gallbladder disease [21]. Because of colitis or caecocolitis, proctosis is the primary clinical symptom when it comes to Helicobacter infection. Animals infected with *Helicobacter* may also develop diarrhea [22].

Park J. M. et al. [23] found that EGCG (epigallocatechin-3-gallate) not only improved various parameters of diabetes mice but also significantly increased the ratio of *Firmicutes*/*Bacteroides* at the phylum level. At the scientific level, EGCG increased the proportion of Christensenaceae and decreased the proportion of *Enterobacteriaceae* and *Proteobacteria*. After Huanglian Jiedu Decoction intervened in fat Zucker rats with diabetes, the proportion of *Firmicutes*/*Bacteroides* was reduced, and the proportion of *Parabacteroides*, *Blautia*, *Akkermannia*, and other bacteria in SCFA-producing and anti-inflammatory bacteria changed [24]. In a randomized clinical trial, a specially designed Chinese herbal formula composed of eight kinds of herbs significantly changed the overall structure of intestinal flora and alleviated the symptoms of type 2 diabetes (T2DM) by enriching *Faecalibacterium*, *Blautia*, and other beneficial bacteria [25,26]. Yasushi Hasegawa et al. found that glucose metabolism was significantly increased in the mantle group compared to the control group, which implied that the mice had elevated fasting blood glucose levels, which could be due to any other form of acute disease or inflammation in the body. This could also severely impair glucose tolerance in the mice, leading to the development of glucose metabolism disorders akin to diabetes. The result of the intestinal flora imbalance caused by SMT is contrary to the conclusion of intestinal flora improvement in diabetic mice in previous studies. Therefore, we hypothesize that SMT disrupts the balance of intestinal flora and damages the structure and composition of intestinal flora, which we guess is one of the causes of glucose metabolism disorder in mice.

The Tukey test (Figure 6d), as shown in the figure, revealed significant differences at the genus level between bacteria, control group, and scallop mantle group. Using LEfSe analysis, it was also found that there were significant differences in microbial communities among the groups (LDA Score > 2) (Figure 6e,f). At the phylum to genus level, it needs to be pointed out that there were 22 different bacteria in the SMT group as compared with only 9 different bacteria in the CON group. By comparison of the SMT group and control group, the bacteria groups are different. Phylum *Proteobacteria*, family *Enterobacteriaceae*, and *Phylum campilobacterota*, class *Campylobacteria*, genus *Helicobacter* significantly enriched in the scallop mantle group.

The variation of taxonomic abundance was consistent with the results of the transition analysis. LEfSe analysis was used to screen for significantly different bacteria between groups. The results indicated that the differences in the control group were mainly non-pathogenic bacteria, while the differences in the SMT group were mainly virulence bacteria, for example, *Escherichia-Shigella* and *Helicobacter*.

### 2.7. Analysis and Prediction of Intestinal Flora Metabolism

The sample data from the control group and scallop mantle group were selected, examined, and predicted on the KEGG platform. The prediction functions of altered microbial communities are mainly involved in (Figure 7); metabolites were predicted, all of which were elevated in the scollop mantle group (*p* < 0.05), including d-Alanine, Vitamin B1, Vitamin B2, Sulfurous acid, L-Asparaginase, Glyoxylic acid, Ubiquinone, Citrate, Aminobenzoate, and Steroid hormone.

Glyoxylate and dicarboxylate metabolism encompasses a series of reactions involving the utilization of glutaric or dicarboxylic acids. Proffitt Ceri et al. [27] discovered an augmented abundance of metabolic reactions related to glutaric and dicarboxylic acids, particularly concerning tartaric acid, in individuals with obesity, type 2 diabetes, and atherosclerosis.

Tartaric acid serves as a metabolite within the glutaric and dicarboxylic acid metabolic pathway. It is commonly present in foods such as grapes, and upon ingestion, it can enter metabolism through either the tricarboxylic acid cycle or conversion into glycerol. Elevated plasma levels of glycerol have been positively linked to type 2 diabetes. Only 20% of ingested tartaric acid from food is excreted in urine, indicating that the remaining portion may be consumed by intestinal microbes. While human tissues are capable of metabolizing tartaric acid, it is predominantly metabolized by intestinal bacteria, highlighting the intimate association between tartaric acid and the intestinal microbiome.

The glyoxylate cycle has the capability to convert fatty acids into glucose, resulting in insulin resistance. During metabolic disorders, the genome-scale metabolic models revealed heightened acetate production, which could be associated with glyoxylate and dicarboxylate metabolisms since the pathways decompose acetate and amino acids in microorganisms for energy generation. There was an enhancement in the arginine and proline metabolism. The enhancement was further correlated with elevated consumption of glutamic acid, which is necessary for the synthesis of arginine and proline. The tartrate undergoes fermentation in the colon. Elevated levels of acetic acid can enhance pancreatic B-cell activity and stimulate insulin secretion, leading to eating habits and obesity that ultimately impact host metabolism. These results suggest that the SMT could disrupt the balance of intestinal microflora and, thereby, cause a disturbance of glucose metabolism. The metabolic pathway of intestinal flora elucidates the disturbance of glucose metabolism in mice caused by scallop mantle toxin found in Yasushi Hasegawa et al.

## 3. Conclusions

This paper investigates the presence of a new type of shellfish toxin on scallop mantle in Dalian, Liaoning Province, China, and the effects of this toxin on the liver and kidney. Additionally, it aims to examine the impact of this new type of shellfish toxin on intestinal flora and intestinal injury. Compared with the CON group, the SMT group mice exhibited decreased body weight and food intake; AST, ALT, γ-GT, TG, TC, TBA, BUN, CRE, and other serum indexes in liver and kidney were increased; the histopathology of liver, kidney, and intestinal tract was seriously damaged; intestinal villi length was reduced; and the concentrations of inflammatory serum indexes such as IL6, TNF-α, and LPS were increased. This study explored how a certain inflammatory response occurs in mice bodies. Regarding the intestinal flora of mice analysis, we found that the beneficial bacterial flora, such as *Muribaculaceae* and *Blautia*, decreased, while the harmful bacterial flora, such as *Escherichia-Shigella* and *Helicobacter*, increased. In this paper, the toxic effects of SMT on intestinal flora were explained from the perspective of intestinal flora. Specifically, SMT caused the disturbance of intestinal flora composition, the decrease in beneficial bacteria, and the increase in harmful bacteria caused intestinal barrier damage and inflammation, ultimately causing toxicity. We are leaning towards believing that there may be a new type of shellfish poison in the mantle of Dalian scallop, and its toxicological mechanism may be similar to that of the mantle of Japan. Continuous consumption of scallop mantle would cause varying degrees of damage to mice livers, kidneys, and intestines, as well as destroy the balance of intestinal flora, eventually resulting in disease.

By integrating the findings of this study with the research progress of Hasegawa’s experimental team, it is found that both mice and rats will suffer certain toxicological damage after consuming the feed added to the scallop mantle, and it is concluded that the scallop mantle contains toxins, which will lead to body death by increasing harmful bacteria in the intestinal flora and reducing beneficial bacteria. Our investigation into this new shellfish toxin is still at a preliminary stage and faces great limitations. For example, we lack knowledge regarding the specific components of this toxin, thereby hindering our ability to develop corresponding antibodies to deal with and treat the damage caused by this toxin. Our future research direction is to establish the separation and purification method of the target proteotoxin, determine the physicochemical properties and molecular structure of the target proteotoxin, and master the rapid and reasonable detection method of the target proteotoxin to elucidate the mechanism of protein toxin formation in scallop skirt. The key factors and techniques to weaken or detoxify the protein toxin through an investigation into the regulation of protein toxin formation so as to solve the safety problem of scallop skirts and promote the utilization of high-value scallop skirts. We seek to standardize the production of processed food for scallop skirt to solve food safety problems.

Our objective is to study the growth conditions of *Pecten yessoensis* and the law of the formation of new shellfish poison, determine whether the new shellfish poison is formed naturally or accumulated by filtering the toxic algae in the ocean, and clarify the difference and relationship between the formation process of new shellfish toxin and the known shellfish poison, as well as the relationship between its formation and the sea area and shellfish species. We also seek to prepare polyclonal antibodies and gene cloning; explore rapid and economical detection techniques; regulate the formation mechanism of new shellfish toxicity by environmental factors, culture methods, sea area, and shellfish species; and explore the key factors and techniques for reducing or removing the virus. In order to further clarify the mechanism of action of the toxin and its impact on human health, we may propose ways to reduce the consumption of shrimp scallop skirt to avoid the harm caused by the new shellfish toxicity to humans so as to not only strengthen the supervision of food safety but also avoid the harm of foodborne diseases.

## 4. Materials and Methods

### 4.1. Collection and Preparation Scallop Mantle Toxin

*Patinopecten yessoensis* scallop samples were purchased from the aquatic products market in Jinzhou, Liaoning Province, China. After that, the seawater was drained, and the scallop mantle was taken out, washed, frozen, freeze-dried, and then ground into powder so that it was ready to add to the mouse diet.

### 4.2. Animal Treatment

Eight-week-old male ICR mice (25–30 g) were purchased from Jinzhou Medical University, Liaoning Province, China. These mice were kept in a circadian cycle with a temperature of 23 ± 2 °C, a humidity of 50 ± 10%, and 12 h of light and 12 h of darkness [28]. After a week of adaptation, the experiment began. All animal experiments followed the guidelines for nursing care and experimental animals of Jinzhou Medical University.

All experimental procedures in this study were in accordance with the guidelines for the use of animal care at the Chinese Academy of Health and have been approved by the Animal Care Use Committee of Animal Center, Jingzhou Medical University, Liaoning Province, China [29], Approval date: 25 October 2023. Use license number: SYXK2022-0006, license number: SCXK2019-0003.

These mice were randomly divided into two (groups 10 mice per group): control group and scallop Mantle group. Control group ate a normal diet; scallop Mantle group ate diets supplemented with scallop mantle. We ensured adequate water supply during breeding, changed the bedding once a week, and weighed once a week. The status of the mice was observed and the amount of remaining feed was recorded [30]. The daily feed is shown in Table 2.

After 4 weeks of starting the feeding and nesting, the mice were anesthetized, and white adipose tissue such as the groin, epididymis, liver, kidney, colon, stomach, and cecum were rapidly removed and weighed. We took heart blood and centrifuged it at 5000× *g* for 10 min. We collected two fecal particles excreted by each mouse in an empty sterile cage and store them at −80 °C for immediate collection until microbiological analysis is conducted.

### 4.3. Biochemical Assays of the Serum

Serum concentrations of total cholesterol (TC), triglyceride (TG) [31], total bile acids (TBA), gamma-glutamyl transpeptidase (γGTP), aspartate aminotransferase (AST), alanine aminotransferase (ALT) [32], creatinine (CRE), blood urea nitrogen (BUN) in liver were measured with the assay kits (Nanjing Jiancheng Institute of Biotechnology, Nanjing, China). All reagents were prepared in accordance with the steps of the kit instruction manual and measured in the enzyme-label instrument and ultraviolet spectrophotometer. The reagents used in this paper were purchased from Nanjing Jiancheng Bioengineering Institute, Nanjing, China. Mouse TNF-α ELISA KIT, Mouse IL-6 ELISA KIT, Mouse LPS ELISA KIT, and the reagents used in this paper were purchased from SHANGHAI JINGKANG BIOENGINEERING Co., Ltd., Shanghai, China. The measurement data were recorded and then analyzed using software.

### 4.4. Histology Examination

After blood extraction, mice were euthanized; kidneys, liver, and intestines were taken in turn; weighed and recorded; and cleaned with PBS solution. Kidney, liver, and intestine tissues were fixed with 4% neuro buffer formaldehyde and inserted parafn according to the procedure. Parafn-embedded kidney, liver, and intestine tissues were cut into slices. Slices were dyed with hematoxylin eosin (HE). We observed the morphology of ileal villi under a microscope. We used ImageJ (K-Viewer-1.7.0.29-x86_64) software to measure the height of villi and the depth of crypts.

### 4.5. Inflammatory Cytokines Examination

The contents of interleukin-6 (IL-6), tumor necrosis factor-α (TNF-α), and Mouse Lipopolysaccharides (LPS) in serum were determined by enzyme-linked immunosorbent assay (ELISA). The serum of mice was collected by ocular blood sampling, and then ELISA was detected in serum at 4 °C centrifuged at 3000× *g* for 10 min according to the instructions of ELISA kit.

### 4.6. 16S Sequencing and Analysis

According to the manufacturer’s instructions, we used GHFDE100 (Zhejiang Hangzhou Equipment Preparation 20190952, Hangzhou, China) DNA Separation Kit (GUHE Laboratories, Hangzhou, China) to extract total bacterial genomic DNA samples from all samples. The quantity and quality of extracted DNAs were measured using a NanoDrop ND-1000 spectros Photometer (Thermo Fisher Scientific, Waltham, MA, USA) and agarose gel electrophoresis, respectively. PCR amplification of the bacterial 16S rRNA genes V4 region was performed using the forward primer 515F and the reverse primer 806R.

After the separate quantification step, equal amounts of amplicons were pooled together for paired-end 2 × 150 bp sequencing using the Illlumina NovaSeq6000 platform at GUHE Info Technology Co., Ltd. (Hangzhou, China) [33,34,35,36,37,38,39,40,41].

### 4.7. Statistical Analysis

SPSS27.0 software package was used to analyze data. All results were presented as mean ± SD, and multiple comparisons of CON and SMT groups were conducted by One-way ANOVA and SNK test. *p* < 0.05 was considered statistically significant [42].

## Figures and Tables

**Figure 1 toxins-16-00247-f001:**
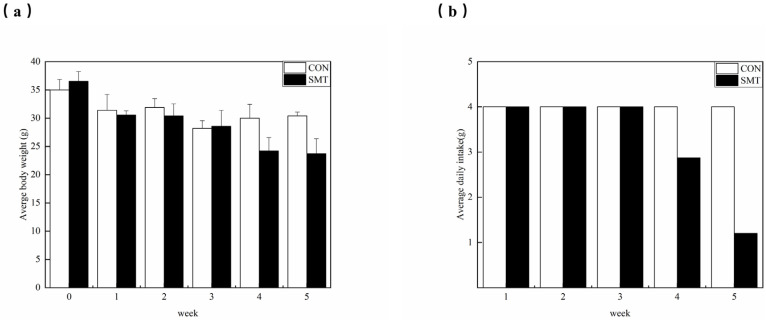
Weekly changes in body weight and food intake. (**a**) is the weekly weight change of CON group and SMT group; (**b**) is the weekly residual feed changes of CON group and SMT group.

**Figure 2 toxins-16-00247-f002:**
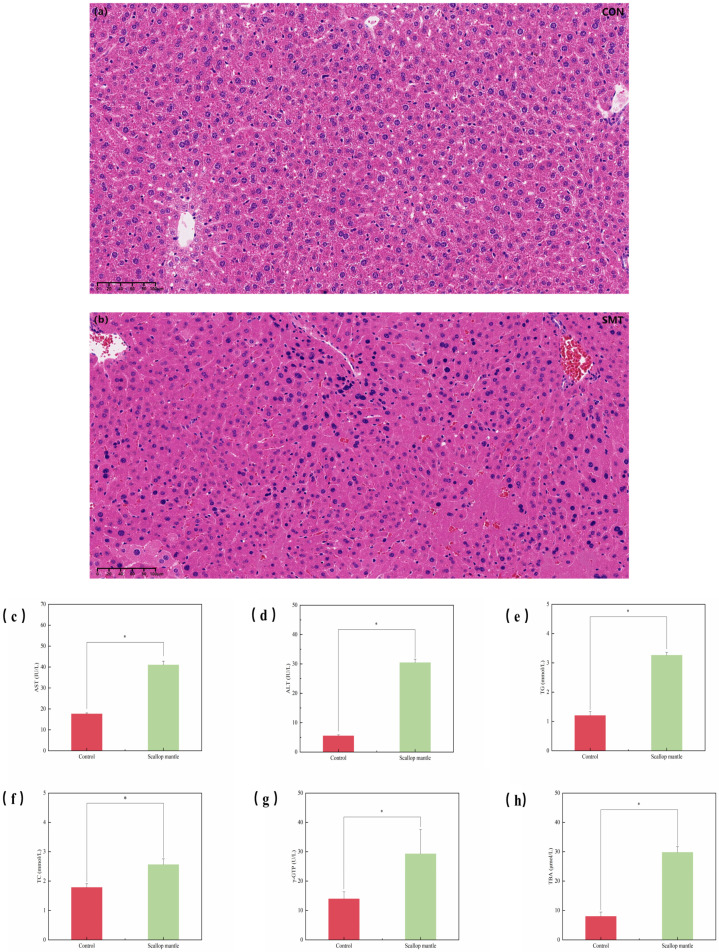
Liver serum markers and histopathology. (**a**) is the liver histopathological picture of CON group; (**b**) is the liver histopathological picture of SMT group; (**c**–**h**) are the serum activity pictures of AST, ALT, TG, TC, γ-GTP and TBA corresponding to CON and SMT groups, respectively.

**Figure 3 toxins-16-00247-f003:**
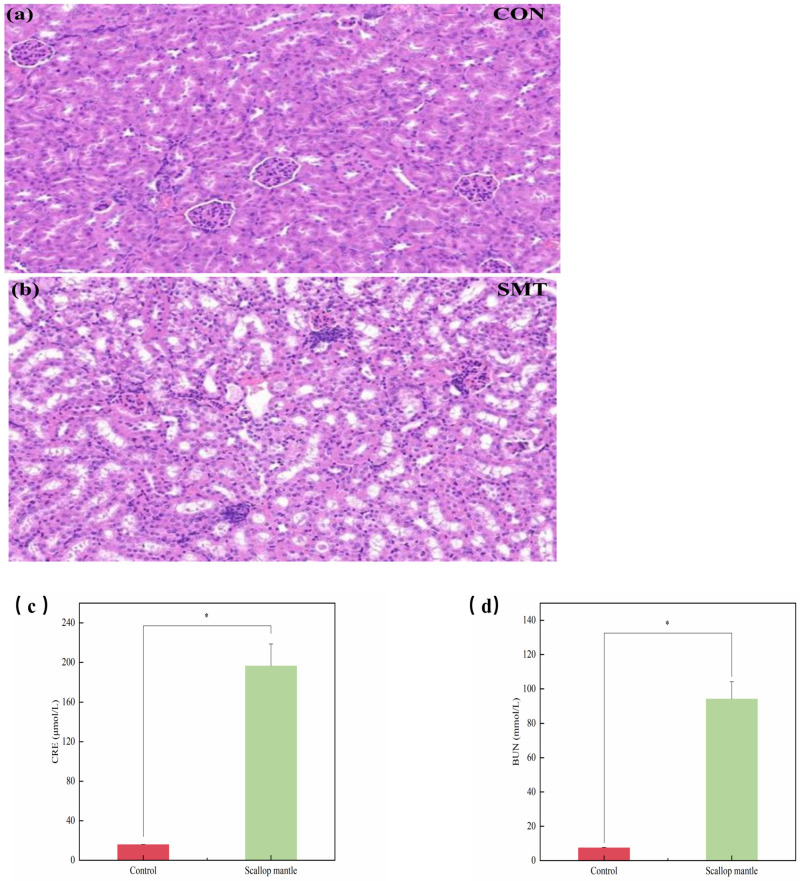
Kidney serum marker and histopathology. (**a**) is the kidney histopathological picture of CON group, (**b**) is the kidney histopathological picture of SMT group, (**c**,**d**) are the serum activity pictures of CRE and BUN corresponding to CON and SMT groups, respectively.

**Figure 4 toxins-16-00247-f004:**
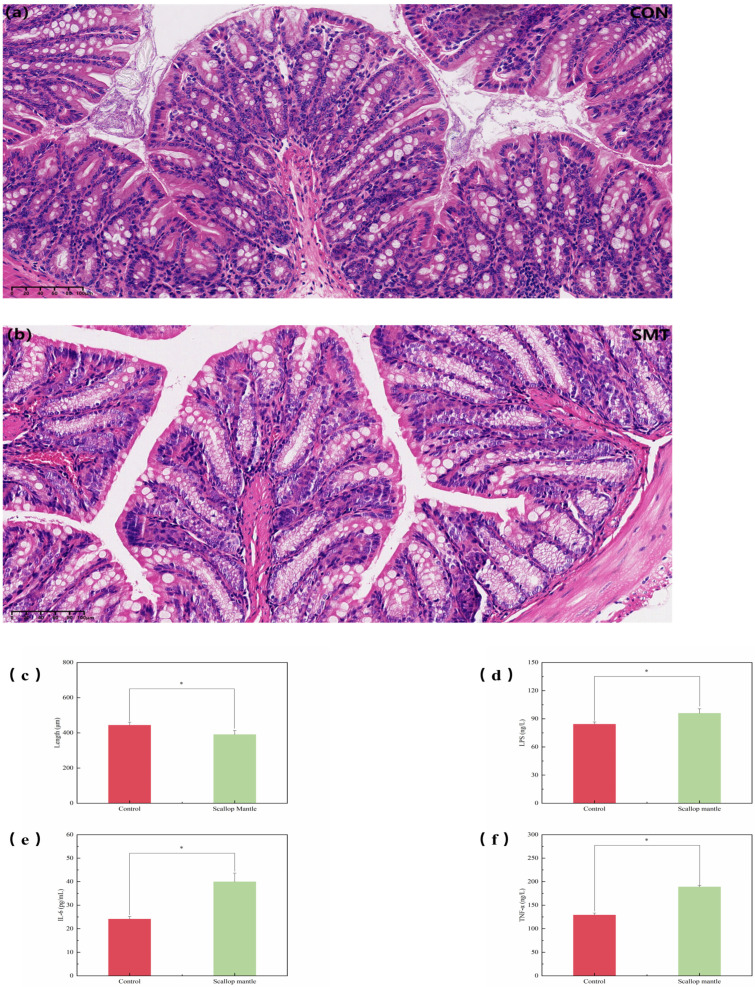
Effect of SMT on colonic histopathological changes and inflammation in mice. (**a**) is the intestinal tract histopathological picture of CON group, (**b**) is the kidney histopathological picture of SMT group, (**c**) is the intestinal villus length of CON group and SMT group, (**d**–**f**) are the serum activity pictures of LPS, IL-6, and TNF-α corresponding to CON and SMT groups, respectively.

**Figure 5 toxins-16-00247-f005:**
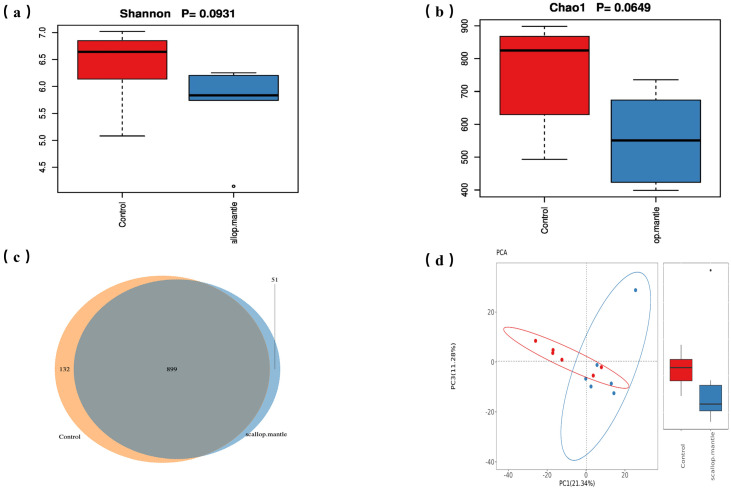
Effects of SMT on intestinal flora diversity in mice. Intestinal flora alpha diversity parameters: observed (**a**) shannon and (**b**) chao1, (**c**) Venn diagram based on OTU levels, (**d**) PCA plot of intestinal flora and b-diversity difference diagram of intestinal flora in two groups.

**Figure 6 toxins-16-00247-f006:**
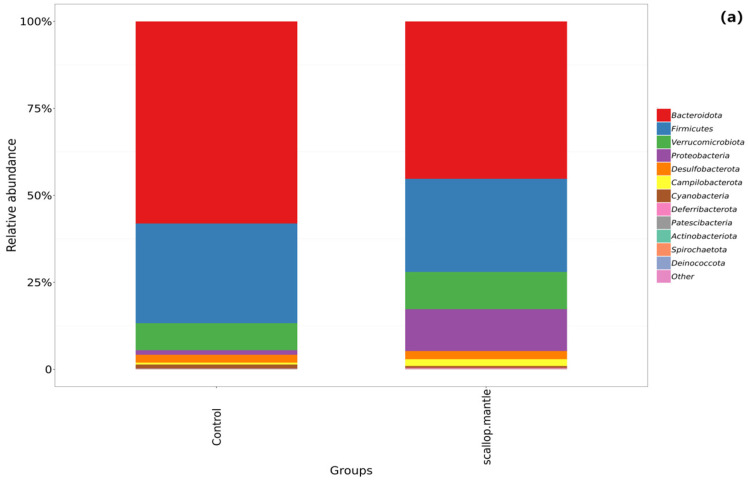
Intestinal flora richness of mice, Tukey test, and LEfSe determinations. (**a**) Species composition histogram at phylum level, (**b**) species composition histogram at family level. (**c**) Species composition histogram at genus level, (**d**) Tukey test—different species between groups, (**e**,**f**) represent Histograms and Cladogram of enriched taxa based on LEfSe determinations, respectively.

**Figure 7 toxins-16-00247-f007:**
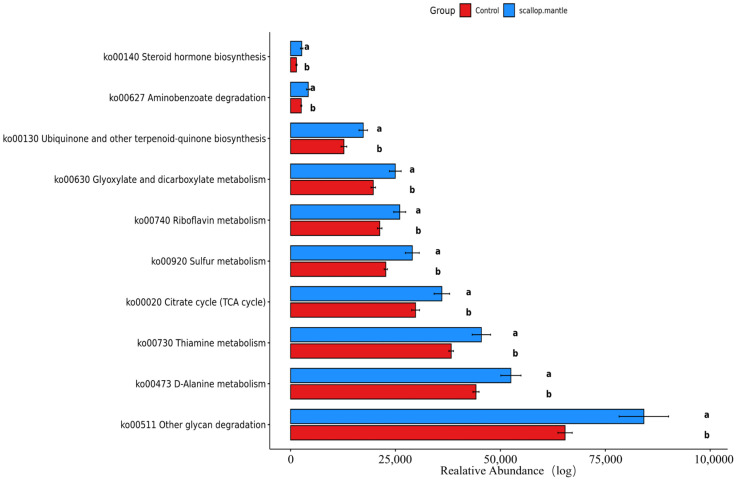
Metabolic prediction pathway of KEGG platform.

**Table 1 toxins-16-00247-t001:** Organ percentage of body weight (organ coefficient %).

	Control	Scallop Mantle
Liver	7.0 ± 0.3	6.5 ± 0.57
Kidney	1.6 ± 0.01	2.0 ± 0.08
Intestinal tract	2.21 ± 0.13	2.24 ± 0.15

**Table 2 toxins-16-00247-t002:** Composition of control diet and mantle diet.

	Control Diet (%)	Scallop Mantle Diet (%)
Casein	20%	20%
Corn starch	15%	15%
Cellulose	5%	5%
Mineral mixture	3.5%	3.5%
Vitamins mixture	1%	1%
Freebase L-cysteine	0.3%	0.3%
Choline bitartrate	0.2%	0.2%
Sucrose	50%	48%
Soybean oil	5%	5%
Scallop mantle	0%	2%
Total	100%	100%

## Data Availability

No new data were created or analyzed in this study. Data sharing is not applicable to this article.

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
