# Peer review of "Effects of Scallop Mantle Toxin on Intestinal Microflora and Intestinal Barrier Function in Mice"

_toxins, 2024, doi:10.3390/toxins16060247_

Round 1
Reviewer 1 Report
Comments and Suggestions for Authors
Author Response
1.What specific toxin(s) were identified in the scallop mantle samples from Dalian, Liaoning Province, China, and how do they compare to known toxins found in scallops from other regions?
Thank you very much for taking the time to review this manuscript.According to the comprehensive research results of Professor Hasegawa's team in Japan and our own research results, we found that a novel protein toxin was identified in the mantle samples of scallops from Dalian, Liaoning Province, China. The isolated toxic substances were protein complexes with molecular weights of 18 kDa and 29 kDa. (Hasegawa Y, Itagaki D, Konno K, et al. Feeding of scallop mantle epithelial cell layer causes subacute toxicity against rodents[J]. Fisheries science, 2018, 84: 91-100.). Known toxins found in scallops from other regions are aralytic shellfish poison, diarrhic shellfish poison, and neurotic shellfish poison it is a low molecular compound. In terms of toxicity, the scallop mantle epithelial cell layer causes toxic changes in the liver, Intestinal and kidney tissues, along with a glucose metabolism disorder, this protein toxin causes Subacute toxicity. aralytic shellfish poison, diarrhic shellfish poison, and neurotic shellfish poison cause acute toxicity.
2.Can you elaborate on the mechanisms by which the scallop mantle toxin(s) exert their toxic effects on the liver, kidney, and intestinal tissues of mice?
Thank you very much for taking the time to review this manuscript.The Aspartate aminotransferase (AST) and Alanine aminotransferase (ALT) of the SMT group mice were significantly increased compared with the control group. ALT is a sensitive indicator of liver damage. ALT mainly exists in the cytoplasm, of which liver cells have the highest content, followed by cardiomyocytes. If liver cells are damaged, ALT will enter the blood, resulting in elevated ALT levels in the blood, and elevated ALT mainly reflects liver cell damage. AST mainly exists in mitochondria, and the highest content is found in cardiomyocytes, followed by hepatocytes. If heart muscle cells or liver cell mitochondria are damaged, AST can enter the bloodstream, causing elevated AST levels in the blood. Compared with the control group, Creatinine (CRE) and Blood urea nitrogen (BUN) of the SMT group mice were significantly increased. CRE and BUN are indicators of kidney function. Higher CRE than normal value is mostly related to kidney damage, and increased BUN may lead to various renal parenchymal lesions. Such as glomerulonephritis, interstitial nephritis, acute and chronic renal failure, intrarenal space occupation and destructive diseases. Compared with the control group, the inflammatory factors Tumor necrosis factor-α (TNF-α), Interleukin-6 (IL-6), Lipopolysaccharides (LPS) were significantly increased, which indicates the inflammatory response of the SMT group mice. The present results suggest that the continuous ingestion of mantle tissue triggers inflammation,ER stress,and oxidative stress in the small intestine,leading to the incorporation of toxic substances into the liver,kidneys,and other parts of the body. In the combined literature (Maeda N,Yumoto T,Xiong G, et al. Characterization and Stability of a Novel Toxin in Scallop Mantle Tissue. Foods. 2023, 12(17): 3224.), it can be shown that the new shellfish poison destroys the intestinal barrier of mice, causing intestinal inflammation, which then leads to systemic inflammatory response, resulting in death. Compared with the control group, the abundance of beneficial bacteria (Muribaculaceae and Blautia) decreased, and harmful bacteria (Escherichia-Shigella and Helicobacter) increased in the SMT group mice. Its increase not only causes the homeostasis of the intestinal flora to be disrupted, resulting in diarrhea, Gastrointestinal diseases. causes varying degrees of acute pathology, ranging from gastroenteritis to chronic pathology, including inflammatory bowel disease and liver and gallbladder disease. Reduced Muribaculaceae richness can lead to a loss of intestinal barrier integrity, which increases the risk of cancer, possibly causing varying degrees of inflammation. A decrease in the abundance of Blautia may lead to health complications such as inflammation and metabolic disorders in the host body. The increase of Escherichia-Shigella richness can lead to the disruption of the dynamic balance of intestinal flora, which can lead to diarrhea, gastrointestinal diseases, etc. Helicobacter can cause varying degrees of acute pathology, ranging from acute gastroenteritis to chronic diseases, including inflammatory bowel disease and hepatobiliary disease due to colitis or enterocolitis. Rectal colitis is the main clinical symptom when infected with Helicobacter pylori, and animals infected with helicobacter pylori may also have diarrhea.
3.Were there any observable differences in the toxic effects of the mantle toxin(s) based on the dosage or duration of exposure in the experimental mice?
Thank you very much for taking the time to review this manuscript.Professor Hasegawa's team found that Mice fed a diet containing 20% mantle tissue did not show a distinct increase in toxicity compared with mice fed a diet containing 1% mantle tissue, demonstrating that feeding mantle tissue does not lead to acute toxicity. References (Hasegawa Y, Itagaki D, Konno K, et al. Feeding of scallop mantle epithelial cell layer causes subacute toxicity against rodents[J]. Fisheries science, 2018, 84:91-100.).
4.In the context of potential human exposure, what are the implications of these findings for seafood safety and public health, particularly in regions where scallops are commonly consumed?
Thank you very much for taking the time to review this manuscript.1. The scallop mantle may be eaten by other Marine organisms, and the protein toxins contained in it will be contaminated by the food chain from the bottom to the top, resulting in pollution of most Marine organisms and serious economic harm. 2. Due to the existence of new shellfish poison, consumers may reduce the purchasing power of scallops, resulting in a certain degree of economic losses. 3. After people eat scallop mantle or other Marine organisms infected with the new shellfish toxin, it may cause damage to the kidney, liver and intestine, or even death, resulting in serious danger to life safety.
5.How do the observed changes in gut microbiota composition and diversity in the SMT group relate to the systemic toxicity and inflammation observed in the mice?
Thank you very much for taking the time to review this manuscript.Compared with the control group, the inflammatory factors Tumor necrosis factor-α (TNF-α), Interleukin-6 (IL-6), Lipopolysaccharides (LPS) were significantly increased, which indicates the inflammatory response of the mice. In the combined literature (Maeda N,Yumoto T,Xiong G, et al. Characterization and Stability of a Novel Toxin in Scallop Mantle Tissue. Foods. 2023, 12(17): 3224.), it can be shown that the new shellfish poison destroys the intestinal barrier of mice, causing intestinal inflammation, which then leads to systemic inflammatory response, resulting in death. Compared with the control group, the abundance of beneficial bacteria (Muribaculaceae and Blautia) decreased, and harmful bacteria (Escherichia-Shigella and Helicobacter) increased. Its increase not only causes the homeostasis of the intestinal flora to be disrupted, resulting in diarrhea, Gastrointestinal diseases. causes varying degrees of acute pathology, ranging from gastroenteritis to chronic pathology, including inflammatory bowel disease and liver and gallbladder disease.The gut microbiota interacts with the host to maintain homeostasis, and imbalances can lead to disease. Firmicutes and Bacteroidetes (F/B) are the most diverse categories that affect the physiological state of human and mouse hosts, and the imbalance of F/B ratio is related to different disease processes. Increasing the increase of F/B ratio will enable intestinal flora to obtain energy more effectively, promote the synthesis of fat and cholesterol, and cause diseases such as hyperlipidemia and lipid metabolism disorders. Compared with the control group , the F/B of the SMT group was increased, indicating the imbalance of intestinal flora in the SMT group, which may be one of the causes of systemic inflammation in mice.Compared with control group ,Reduced Muribaculaceae richness can lead to a loss of intestinal barrier integrity, which increases the risk of cancer, possibly causing varying degrees of inflammation. A decrease in the abundance of Blautia may lead to health complications such as inflammation and metabolic disorders in the host body. The increase of Escherichia-Shigella richness can lead to the disruption of the dynamic balance of intestinal flora, which can lead to diarrhea, gastrointestinal diseases, etc. Helicobacter can cause varying degrees of acute pathology, ranging from acute gastroenteritis to chronic diseases, including inflammatory bowel disease and hepatobiliary disease due to colitis or enterocolitis. Rectal colitis is the main clinical symptom when infected with Helicobacter pylori, and animals infected with helicobacter pylori may also have diarrhea.These major microbiota changes may be one of the causes of systemic toxicity and inflammation
- Have similar studies been conducted on other shellfish species or marine organisms to compare the toxicity profiles and potential risks associated with consuming different types of seafood?
Thank you very much for taking the time to review this manuscript.In addition to the scallop there are the following species of scallop mantle,figure of body weight change in mice。We only studied animal experiments (body weight changes), and the skirt samples were from Jinzhou City, Liaoning Province, China, including control group, ark shell group, Azumapecten farreri group, Ostreidae group, Mactra veneriformis group. After 7 days of adaptive feeding, mice were weighed, randomly divided into 5 groups with 3 mice in each group, and fed the diet containing 2% samples. One mice died in the ark shell group, two mice died in the Azumapecten farreri group, and one mouse died in the Ostreidae group, but the cause of death has not been identified and needs further study. but whether the cause of death is toxins, the presence of toxins, needs further research.
7.Are there any identifiable biomarkers or early indicators that could be used to detect and monitor the onset of toxicity from scallop mantle consumption in both experimental models and potentially in human populations?
Thank you very much for taking the time to review this manuscript.At present, the progress of our research is that we have not isolated this protein toxin and have not conducted amino acid sequence determination, so there are no identifiable biomarkers or early indicators that can be used to detect and monitor the occurrence of scallop mantle toxicity in experimental models and potential populations. At present, we are using animal experiments to verify its toxicity.
- What steps are being taken or recommended to mitigate the potential risks associated with scallop mantle toxins in seafood production and consumption?
Thank you very much for taking the time to review this manuscript.1. Publicize our research progress to aquatic products consumers, including the subacute toxins produced after mice eat scallop mantle, which will cause certain damage or even death to liver, kidney and intestine, and may cause the same result to humans. 2. Reduce the intake of scallop mantle. This protein toxin is accumulated for a long time and cannot be eaten for a long time. 3. The methods to alleviate the toxin are still under research, such as heating, chemical, bleaching, acid-alkalization, etc., can not reduce the effect of the toxin, and further research is needed.
- Considering the observed damage to the intestinal mucosal barrier, have there been investigations into potential therapeutic interventions or protective measures to prevent or reverse these effects?
Thank you very much for taking the time to review this manuscript.At present, 1. Reduce the interference of intestinal barrier oxidative stress. For example, some foods with good antioxidant properties contain polysaccharides, polyphenols and other ingredients to interfere, and improve their inflammation and antioxidant indexes to protect or counter the damage caused by toxins to intestinal barrier. 2. The new shellfish poison will cause the increase of harmful bacteria and the decrease of beneficial bacteria, but some foods and probiotics will reduce harmful bacteria and increase beneficial bacteria, so that the adverse effects caused by it can be reversed.
10.What further research avenues are planned or suggested to deepen our understanding of the toxicological mechanisms, ecological implications, and broader impacts of scallop mantle toxins on marine ecosystems and human health?
Thank you very much for taking the time to review this manuscript.1.Establish the separation and purification method of the target proteotoxin(ion-exchange column, salt-out precipitation method, gel filtration column, hydrophobic chromatography), determine the physicochemical properties and molecular structure of the target proteotoxin, and master the rapid and reasonable detection method of the target proteotoxin. The isoelectric point, molecular weight, primary structure and secondary structure of the protein were analyzed by instruments and equipment, and the mechanism of action of the protein toxin was further discussed after a comprehensive analysis of the protein toxin. Polyclonal antibodies were prepared from antigen-immunized rats, and the specificity and targeting of the antibodies were identified by Western blot method, which provided a theoretical basis for the detection or quantification of targeted toxins. 2. To elucidate the mechanism of protein toxin formation in scallop skirt. The key factors and techniques to weaken or detoxify the protein toxin were found out by studying the regulation of protein toxin formation, so as to solve the safety problem of scallop skirt and promote the utilization of high value of scallop skirt. Standardize the production of processed food for scallop skirt to solve food safety problems.
Reviewer 2 Report
Comments and Suggestions for Authors
This manuscript shows potential for publication in Toxins, but needs revisions to address the drawbacks mentioned below.
In the abstract, briefly mention the toxin source. Highlight the key findings: intestinal damage, gut microbiota changes, and potential glucose metabolism disorder. Emphasize the significance for human health: potential food safety risk.
In the introduction, concisely mention previous research on scallop toxins. Focus on the rationale for investigating the toxin’s effects on the gut microbiota and glucose metabolism.
Include statistical details (tests used, p-values) for all comparisons mentioned in the text in the results section.
Strengthen the discussion by integrating the findings with previous research on scallop toxins and gut microbiota. Also discuss the limitations of the study. Propose future research directions to further elucidate the mechanism of action of the toxin and its implications for human health.
Briefly summarize the key findings and their significance for food safety. The conclusions drawn from the results should be directly linked to the objectives and hypotheses of the study.
The manuscript requires thorough language editing for grammar, clarity, and scientific tone. Consider using a professional editing service.
Generally, this manuscript has a good foundation, but revisions are necessary to meet the journal’s standards for clarity, impact, and scientific language.
Comments on the Quality of English LanguageThe manuscript requires thorough language editing for grammar, clarity, and scientific tone. Consider using a professional editing service.
Author Response
1.Strengthen the discussion by integrating the findings with previous research on scallop toxins and gut microbiota. Also discuss the limitations of the study. Propose future research directions to further elucidate the mechanism of action of the toxin and its implications for human health.
Briefly summarize the key findings and their significance for food safety. The conclusions drawn from the results should be directly linked to the objectives and hypotheses of the study.
Thank you for pointing this out. We agree with this comment.
In response to your question, we've highlighted the answers in blue in the conclusion.The specific location is in blue font in the second and third paragraphs of 3.Conclusion on page 13.
2.The manuscript requires thorough language editing for grammar, clarity, and scientific tone. Consider using a professional editing service.
Thank you for pointing this out. We agree with this comment.We are trying to fix the bad syntax.For some syntax errors, we have made the corresponding changes with blue marks.The location is indicated in blue on pages 1,2,4,7,8,9,10,11,12,14.
Reviewer 3 Report
Comments and Suggestions for Authors
Dear authors
Happy day
The paper unfortunately need a lot of adjustment. Kindly following my comments.
1- Kindly respect the scientific writing general roles. For example the scientific names must be italic. Kindly found that through all the text.
2- You use USA style in the writing, kindly be sure that all the text have the same style.
3- The county names must start with capital letter.
4- If you decided to write a name with capital, so do that in the entire text and the same if you decided to stat it with small letter.
5- Sentences must end with point or the like.
6- In the figure(s), kindly use arrows to spot what you need to highlight.
Some images are not clear and need more contrast and more space.
6- Be sure that you are following the journal style. For example in the text is that correct to cite a reference like "Molecular Medicine Reports, 2016, 13(4): 3052-3062. et al.". And Park J. M. et al [23] .
7- More references are needed.
8- Put the "Conflicts of Interest: " section alone in correct title format.
9- It is better to write the full name of the abbreviated words such as "AST, ALT, TG, TC, ?-GTP, TBA". And make a list of abbreviation.
10- In sentences like "Upon intestinal flora of mice analysis, we found that, the beneficial bacterial flora such as Muribaculaceae and Blautia decreased" did you know the species name. Or why you did not use the species name?
11- In the results section you describe Helicobacter as a harmful bacteria, but in the conclusion you describe it as "the beneficial bacterial flora such as Escherichia, Shigella and Helicobacter increased". I am sure that it is a typing mistake. If not so kindly be sure from each single information you write!
This is a part of the mistakes I found in a fast read for you document and I am recommended major revision and including all my comments.
With my pleasure

Author Response
1- Kindly respect the scientific writing general roles. For example the scientific names must be italic. Kindly found that through all the text.
Thank you for pointing this out. We agree with this comment.We have changed the scientific name in red italics on pages 1,2,8,9,10,12.
2- You use USA style in the writing, kindly be sure that all the text have the same style.
3- The county names must start with capital letter.
4- If you decided to write a name with capital, so do that in the entire text and the same if you decided to stat it with small letter.
5- Sentences must end with point or the like.
Thank you for pointing this out. We agree with this comment.We have corrected these errors accordingly in red font on page 3,4,9,12,13,14 of the article.
6- In the figure(s), kindly use arrows to spot what you need to highlight.
Some images are not clear and need more contrast and more space.
Thank you for pointing this out. We agree with this comment.We have updated these pictures accordingly, specifically in the pictures on page 3,5,6,7,8,11 of the article.
8- Put the "Conflicts of Interest: " section alone in correct title format.
Thank you for pointing this out. We agree with this comment.We have updated the corresponding Conflicts of Interest in red font 6. Conflicts of Interest on page 16 of the paper
9- It is better to write the full name of the abbreviated words such as "AST, ALT, TG, TC, ?-GTP, TBA". And make a list of abbreviation.
Thank you for pointing this out. We agree with this comment.We have updated the English Abbreviation accordingly, and the specific position is in the red font of the 5. Abbreviation English comparison table on page 15 of the paper.
10- In sentences like "Upon intestinal flora of mice analysis, we found that, the beneficial bacterial flora such as Muribaculaceae and Blautia decreased" did you know the species name. Or why you did not use the species name?
Thank you for pointing this out. We agree with this comment.After consulting relevant literature, we did not find other names of intestinal flora Muribaculaceae and Blautia, we hope you can forgive us.
11- In the results section you describe Helicobacter as a harmful bacteria, but in the conclusion you describe it as "the beneficial bacterial flora such as Escherichia, Shigella and Helicobacter increased". I am sure that it is a typing mistake. If not so kindly be sure from each single information you write!
Sorry reviewers, due to our carelessness, harmful bacteria was written as beneficial bacteria. We have made corresponding changes to this mistake, and the specific position is in the red font of 3. Conclusion on page 12 of the article.
Finally, thank you for taking time out of your busy schedule to review the manuscript for us. Thank you very much!
Round 2
Reviewer 2 Report
Comments and Suggestions for Authors
The manuscript has been edited for language and scientific quality. However, a few minor grammatical errors were still spotted, which can easily be rectified during the proofreading of the final galley, if the manuscript is accepted.
Comments on the Quality of English LanguageThe manuscript has been edited for language and scientific quality. However, a few minor grammatical errors were still spotted, which can easily be rectified during the proofreading of the final galley, if the manuscript is accepted.
Reviewer 3 Report
Comments and Suggestions for Authors
Dear Authors
Many thanks for all the correction you made.
The paper now is fin.
